# Magnetic Resonance Imaging during Proton Therapy Irradiation Allows for the Early Response Assessment of Pediatric Chordoma

**DOI:** 10.3390/diagnostics11061117

**Published:** 2021-06-18

**Authors:** Sabina Vennarini, Dante Amelio, Stefano Lorentini, Giovanna Stefania Colafati, Antonella Cacchione, Rita De Vito, Andrea Carai, Benedetta Pettorini, Maurizio Amichetti, Angela Mastronuzzi

**Affiliations:** 1Proton Therapy Center, Hospital of Trento, Azienda Provinciale per I Servizi Sanitari (APSS), 38123 Trento, Italy; dante.amelio@apss.tn.it (D.A.); stefano.lorentini@apss.tn.it (S.L.); maurizio.amichetti@apss.tn.it (M.A.); 2Oncological Neuroradiology Unit, Department of Imaging, Bambino Gesù Children’s Hospital, IRCCS, 00165 Rome, Italy; gstefania.colafati@opbg.net; 3Department of Paediatric Haematology/Oncology, Cell and Gene Therapy, Bambino Gesù Children’s Hospital, IRCCS, 00165 Rome, Italy; antonella.cacchione@opbg.net (A.C.); angela.mastronuzzi@opbg.net (A.M.); 4Histopathology Unit, Bambino Gesù Children’s Hospital, 00165 Rome, Italy; rita.devito@opbg.net; 5Neurosurgery Unit, Department of Neuroscience and Neurorehabilitation, Bambino Gesù Children’s Hospital, IRCCS, 00165 Rome, Italy; andrea.carai@opbg.net; 6Paediatric Neurosurgery Department, Alder Hey Children’s Hospital, Liverpool 00165, UK; benedetta.pettorini@alderhey.nhs.uk

**Keywords:** chordoma, proton therapy, response assessment, pediatric

## Abstract

Chordoma in pediatric patients is very rare. Proton therapy has become a gold standard in the treatment of these neoplasms, as high dose escalation can be achieved regarding the target while maximizing the sparing of the healthy tissues near the tumor. The aim of the work was to assess the evolution of morphological sequences during treatment using T1/T2-weighted magnetic resonance imaging (MRI) for the early response assessment of a *classic chordoma* of the skull base in a pediatric patient who had undergone surgical excision. Our results demonstrated a significant quantitative reduction in the residual nodule component adhered to the medullary bulb junction, with an almost complete recovery of normal anatomy at the end of the irradiation treatment. This was mainly shown in the T2-weighted MRI. On the other hand, the *classic* component of the lesion was predominantly present and located around the tooth of the axis. The occipital condyles were morphologically and dimensionally stable for the entire irradiation period. In conclusion, the application of this type of monitoring methodology, which is unusual during the administration of a proton treatment for chordoma, highlighted the unexpected early response of the disease. At the same time, it allowed the continuous assessment of the reliability of the treatment plan.

The patient was a 10-year-old female with a severe occlusion in the upper airways (highlighted by an urgent first MRI) due to a bulky expansive lesion with net margins involving nearly the whole clivus area, extending up to C2–C3. The lesion studied in the morphological T1/T2-weighted sequences presented hyperintense, non-homogenous characteristics in T2 (see Figure 1b,c) and an isointense appearance in T1 (see Figure 1a), with intense enhancement after administration of the contrast medium.

Given the complexity of the case due to the location and extension of the disease, three surgical operations were performed: two via an endoscopic approach (via oral pathway) and one defined as open. All surgeries confirmed the histologic diagnosis of a *classic chordoma* (see Figure 2c).

Chordomas are a rare disease; they represent 0.2% of primary brain tumors and fewer than 5% of primary bone tumors. About 5% of all the chordomas tumors occur in patients under 20 years of age. They originate from the primitive residues of the notochord, with an average age at diagnosis of 10 years and a male-to-female ratio equal to 1 [1,2].

Pediatric chordomas are most frequent in the intracranial area, with a typical localization in the sphenoidal-occipital synchondrosis and a more common “undifferentiated” histology [3,4].

MRI provides a perfect tool for the evaluation of the tumor mass and the surrounding vital organs (brainstem and cervical medulla), as well as the vascular system and the cranial nerves.

The *classic chordoma* in MRI is iso/hypointense in the T1-weighted images but typically hyperintense in the T2-weighted images, this being due to their increased water component. This can also be associated with mucous, hemorrhages, and calcification. Lobular aspects and multiple hypo-intense septa that are connected to areas of necrosis or cartilage were present in this case.

The reaction to gadolinium is extremely heterogeneous and can vary from homogenous to marked enhancement (with a honeycomb like appearance and linear areas of non-enhancement) [5,6].

Figure 2 “a” and “b” show the axial T2-weighted MRI with the macroscopic residual of the disease post-surgery, just before the beginning of proton therapy treatment.

As can be seen in the top image, the disease was present bilaterally on the tooth of the axis and was characterized by two different MRI-signaled components in T2, with one component being frankly hyperintense in T2, typical of the radiological signal of the *classic* histology upon the right condyle (thin red arrow in Figure 2a,c).

One nodule component was non-homogenous hypo–hyperintense in T2 and was represented by a left para-medullary bulb with compression and a shift on the spinal cord (thick red arrow in the Figure 2b); such a feature is less common in the *classic* type of histology but is more typically present in *de-differentiated* chordomas (it is possible that it can appear in post-surgical resection) [7,8]. The histological picture is of a neoplasm composed by chords and strands of tumor cells embedded in a myxoid background. The tumor cells show abundant pink cytoplasm and round regular nuclei with little cytological atypia. Some cells show multiple cytoplasmic vacuoles creating the classic bubbly appearance of physaliferous cells. On immunohistochemical examination, the neoplasm appears diffusely immunoreactive for cytokeratin and EMA and shows nuclear immunoreactivity for brachyury and variable S100 positivity. A diagnosis of conventional chordoma was made.

The surgeon implanted titanium spinal stabilizers and a nasogastric tube for feeding. The patient was in fair psychological and clinical conditions before the start of proton therapy. There were motor and sensory neurological deficits of the following cranial nerves: IX–X–XI–XII. Paralysis of the left vocal cord and difficulty in completely opening the oral cavity were evident, as well as severe oropharyngeal dysphagia for solid and liquid foods.

The proton therapy intent was curative, and was performed at the Proton Therapy Centre in Trento, Italy.

Despite the slow growth rate, the prognosis for this kind of neoplasia is poor. Currently, the local recurrence rate remains high, being up to 20% in the first-year post diagnosis. Unfortunately, the prognostic factors for chordoma are mostly unknown [5,9]. Currently, the first treatment choice is surgical excision. Resections are unlikely to be complete and adjuvant radiotherapy is administered in most of the cases.

Nowadays, proton therapy lends itself to being the gold standard in the treatment of pediatric chordoma [10,11].

The need for high dose gradients to keep the disease locally controlled, as well as the presence of vital organs at risk (OARs) of severe complications, determined the choice of proton therapy as the radiotherapeutic treatment option. The global survival rate reported in the literature is equal to 89% and 60%, respectively, at 5 and 7.25 years from diagnosis in patients with chordomas of the skull base who undergo surgery and proton therapy [2,12].

The side effects of proton therapy in these published cases may seem minor with respect to those published for conventional radiotherapy. This only reinforces the dosimetric potential of proton therapy to reach high dose gradients while safeguarding nearby OARs.

The main side effects are represented by hypopituitarism, sensorineural hearing loss, worsening of visual impairment, temporal lobe necrosis, and neurological damage due to brain stem and spinal cord injury [13].

A computed tomography (CT) scan of the patient in supine position was acquired. To guarantee the immobilization and reproducibility of the patient position on the treatment couch, a thermoplastic mask coupled with a customized cushion made of a dedicated foam-like material was used. The CT scan and the pre-treatment MRI images were registered to help the clinician to identify the structures of interest. The clinician used the CT images to define the target volume and OARs to be spared. The treatment plan was then prepared using a commercial treatment planning system (Elekta XIO) equipped with a pencil beam dose calculation algorithm. A 3-field (1 posterior and 2 anterior-oblique) plan was created using a single-field-optimization technique [14]. The total prescription dose for the target was 73.8 GyRBE, delivered in 41 fractions (1.8 GyRBE per fraction, 5 fractions a week), while the main constraints (expressed as allowed dose to 1% of the volume) were set for the brainstem and spinal cord at 54 GyRBE and 50 GyRBE, respectively. The treatment was delivered in a proton therapy gantry room featuring a pencil beam active scanning technique. The Figure 3 shows the planned dose distributions on the CT scan. 

The following contours are reported: planning target volume 0–54 GyRBE (blue), planning target volume 54–73.4 GyRBE (red), nodule component (green), brainstem (purple), and spinal cord (yellow).

Four MRI examinations (T2-weighted sequence, without contrast) were acquired over the treatment course at doses of 32.4, 54, 64.8, and 73.8 GyRBE, respectively (Figure 4). 

The figure shows temporal sequence of T2-weighted MR axial images of the nodular component at different times. The upper-left image represents the situation at diagnosis, while the others (clockwise direction) show the evolution during the treatment.

The bony component of the lesion around the tooth of the axis did not change in volume or morphology over the course of treatment, whereas the nodule component showed a progressive dimensional modification from 2.323 cc at the beginning of the treatment to 2.829 cc at 32.4 Gy RBE, 2.051 cc at 54 Gy RBE, and 1.045 cc at 64.8 Gy RBE and 73.4 Gy RBE (Figure 5). This led to a total reduction from the beginning to the end of proton therapy of 1.278 cc (about 55% of the initial volume). This reduction allowed the decompression of the spinal cord.

The early response of this nodular component of chordoma was an unexpected finding during the irradiation, which should be described as an early therapeutic response. Such a result is even more meaningful considering that this tumor area received less than prescribed dose because of its proximity to the spinal cord. At the same time, although the dose to the spinal cord was constrained to 54 GyRBE, thanks to the quality of the treatment plan, at least 50% of this tumor area received 70 GyRBE. This may justify the good radiological outcome.

The irradiation with proton therapy at high dose levels in a child with such a lesion was not trivial. Taking into account the critical OARs like the cervical chord, bulb-cervical chord junction, brainstem in close proximity to the tumor, and the dose gradient required to cover the target on one hand while sparing the OARs on the other hand makes this kind of treatment very challenging. The radiological monitoring, due to the unexpected early response of the nodular component, allowed us to check that the irradiation took place in a safe manner without damaging the critical structures. Despite the dimensional changes in the nodular components adjacent to the bulb-cervical cord junction, the margin adopted for planning proved to be adequate, so no adaptive re-planning was performed.

The available literature describes how these lesions alter little or do not change in size. However, they do remain stable after irradiation regardless of whether conventional and/or particle (protons and ions) therapy is used [15].

This diagnostic and clinical finding highlighted using MRI radiological monitoring during proton treatment in a pediatric patient with chordoma in the skull base may represent a new approach to clinical radiological monitoring. This approach provides an advantage to ensure the reliability of the radiation treatment, providing data with regard to plan evaluation and the safety of the patient.

## Figures and Tables

**Figure 1 diagnostics-11-01117-f001:**
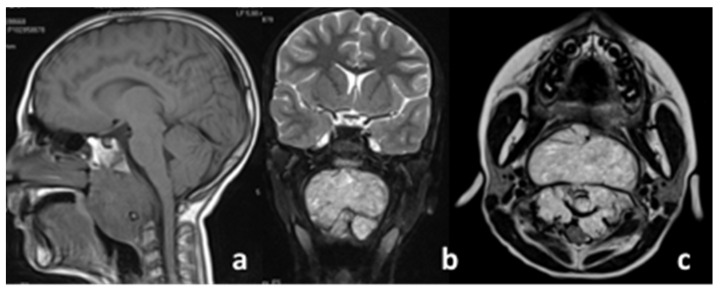
Imaging at diagnosis: (**a**) sagittal view, (**b**) coronal view, (**c**) axial view.

**Figure 2 diagnostics-11-01117-f002:**
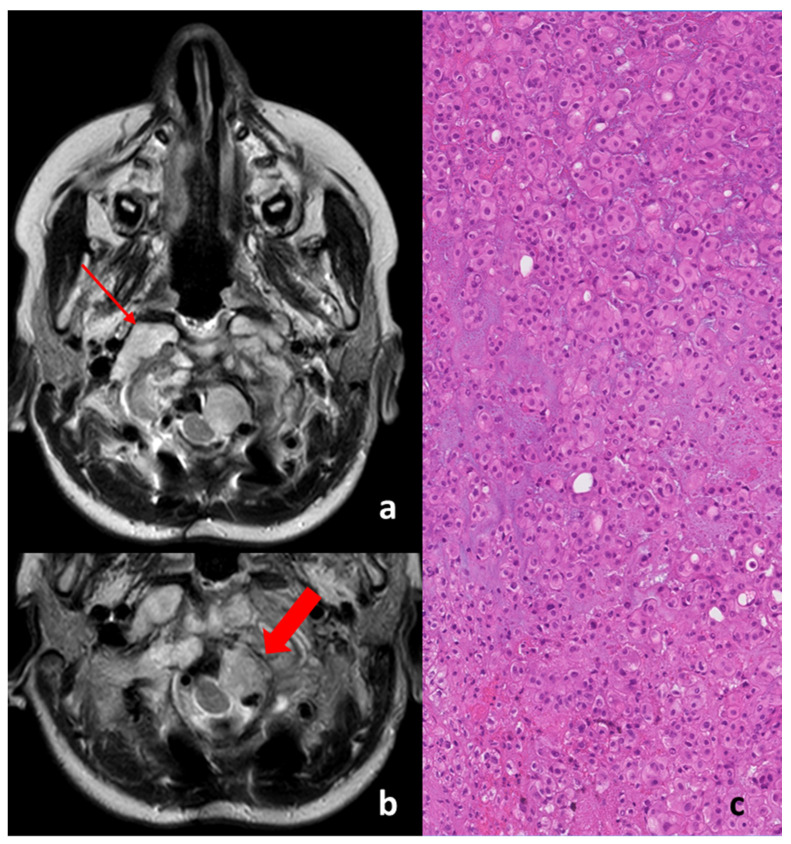
Pre-proton therapy images (representative T2-weighted MRI axial views (**a**) and (**b**)) and histologic examination after biopsy at the first diagnosis (**c**).

**Figure 3 diagnostics-11-01117-f003:**
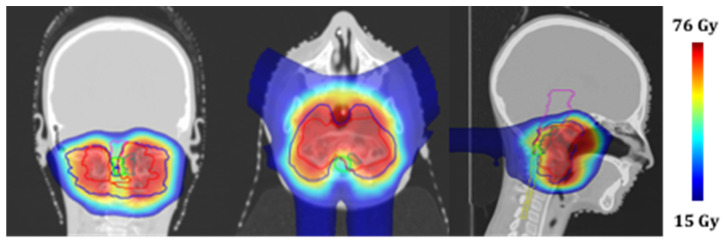
Treatment planning: proton dose distributions in coronal, axial and sagittal views.

**Figure 4 diagnostics-11-01117-f004:**
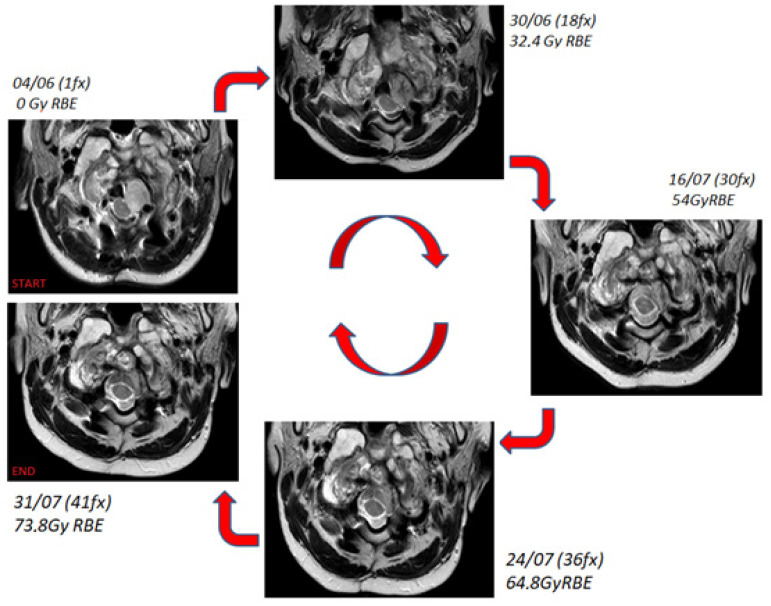
Treatment monitoring over the proton therapy course with T2-weighted MRI.

**Figure 5 diagnostics-11-01117-f005:**
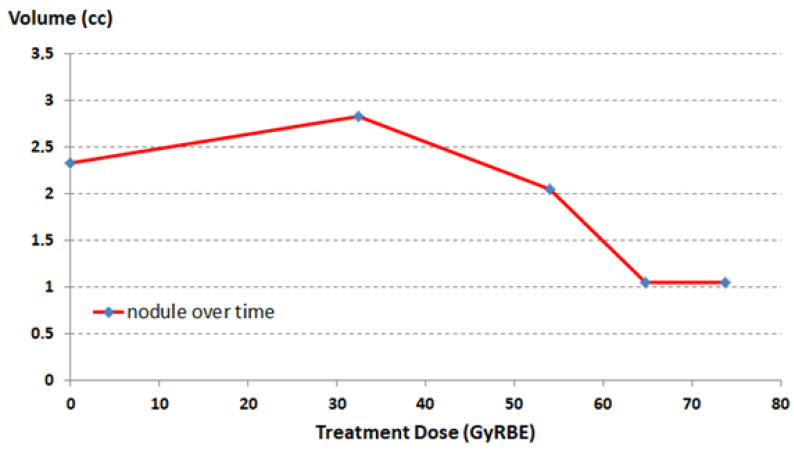
The plot shows the nodule component reduction over proton treatment course.

## Data Availability

Not applicable.

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
