# Peer review of "Magnetic Resonance Imaging during Proton Therapy Irradiation Allows for the Early Response Assessment of Pediatric Chordoma"

_diagnostics, 2021, doi:10.3390/diagnostics11061117_

Round 1

Reviewer 1 Report

Thank you for this interesting paper which looks at the monitoring of  patient with a chordoma during proton beam therapy. Whilst this is interesting I have number for questions which I  feel should be answered in the paper. 

  1. Did you have histological specimens form both areas of differential response mentioned and di they differ.
  2. Do you have any more detailed molecular analysis of the samples?
  3. The regular monitoring with detailed scan during proton beam radiation is unusual, was there  a particular clinical reason for doing this or was this a  research  protocol.
  4. You fail to discuss why apart from interest this has clinical utility to become standard practice or even if research what help would it offer clinicians treating these patients. Did you alter your treatment plan mid treatment on the psi of test findings.

Author Response

Thanks to the reviewer for his/her interesting suggestions. Hereafter a point-by-point reply.

Reviewer #1:

Thank you for this interesting paper which looks at the monitoring of  patient with a chordoma during proton beam therapy. Whilst this is interesting I have number for questions which I  feel should be answered in the paper. 

1) Did you have histological specimens from both areas of differential response mentioned and did they differ.

No we didn’t, biopsies were performed at the same site.

2) Do you have any more detailed molecular analysis of the samples?

No molecular investigations have been carried-out.

3) The regular monitoring with detailed scan during proton beam radiation is unusual, was there  a particular clinical reason for doing this or was this a  research  protocol.

We confirm that the regular radiological monitoring over the treatment course for chordoma cases is not standard in the clinical routine, both in proton and in photon radiotherapy. The implementation of an imaging protocol (mainly using T2-weighted sequences of MRI) during the proton therapy for this specific case was driven by clinical reasons, so it cannot be considered a research protocol. The use of T2-weigheted MRI sequences, being the chordoma a disease originating from the notochord and being rich of water,  allowed us to visualize any possible morphological/dimensional modifications of the lesion in case they would occur. For this young child indeed any possible modification would be critical from a clinical point of view considered that a compression at the level of bulb-cervical chord junction was already present prior the irradiation beginning.   

4) You fail to discuss why apart from interest this has clinical utility to become standard practice or even if research what help would it offer clinicians treating these patients. Did you alter your treatment plan mid treatment on the psi of test findings.

The clinical utility of implementing a radiological monitoring over the proton therapy course using MRI for a chordoma disease located in a critical site, as the one reported here, is, in our opinion, extremely interesting and could be of example for setting new work methods, but also for setting research protocols applied to the clinical practice. Hereafter a brief summary to explain the motivation:

- In literature very few data are currently available about how chordomas behave over the radiotherapy course, both for proton and photon therapy. The number of paeditric series available is small and most of them are represented by case reports which show as the chordoma is morphologically stable after radiotherapy when investigated with radiological imaging. Few are the cases, only adults, where pseudo-progression after radiotherapy is reported. In that respect our specific case is interesting because it shows that a chordoma can be not morphologically stable, probably due to the histologic components of different nature, confirmed by the different radiological signal. So, in critical clinical conditions, like the one reported, this kind of monitoring, implemented for the continuous assessment of the bulb-cervical chord junction allowed to a) deny (for this specific case) the stability of the morphological findings as reported by the scarce literature, b) to surprisingly have a criterion for an early clinical response for one of the two lesion components with evidence of no complications for the child.  

- Furthermore we do believe that the irradiation with proton therapy at high dose levels of a child with such a lesion is not trivial. Knowing as crucial organs at risk like cervical chord, bulb-cervical chord junction, brainstem are in close proximity with tumor and the dose gradient required to cover the target (73.8 Gy RBE for target coverage) on one hand and sparing the organs at risk (54Gy RBE as maximum dose required) on the other hand makes this kind of treatment very challenging. The radiological monitoring, due to the unexpected early response of the nodular component, allowed us to check that the irradiation took place in a safe manner, without damaging the critical structures. 

- Despite the dimensional changes of the nodular components adjacent to the bulb-cervical cord junction, the margin adopted for planning proved to be adequate, so no adaptive re-planning was performed.

Reviewer 2 Report

This article entitled "Magnetic Resonance Imaging during proton therapy irradiation allows early response assessment of a paediatric chordoma" provides a case report of MRI imaging demonstrating lesion changes during the proton irradiation course of a patient. 

There are a couple of areas needing correction/clarification:

1) "Paediatric chordoma is a rare disease; they represent about 5% of all tumours in patients under 20 years of age."

This is incorrect and far exceeds incidence. 

2) Treatment is noted to be single field optimization, however posterior beam shows heterogeneity across the volume. Please ensure that this is not multi-field optimization.

3) Mixed radiographic response, authors note bony component stable and nodular component decreasing. They both appear to be emanating from the clivus with extraosseous extension or expansion. Is there any thought/rationale to this variable response?

4) Can authors comment on actual tumor coverage with prescription dose? Presumably the nodular lesion actually received less dose than the other regions given noted dose constraints. 

Author Response

Thanks to the reviewer for the interesting suggestions/comments. Herafter we reply point-by-point to his/her questions.

Reviewer #2:

This article entitled "Magnetic Resonance Imaging during proton therapy irradiation allows early response assessment of a paediatric chordoma" provides a case report of MRI imaging demonstrating lesion changes during the proton irradiation course of a patient. 

There are a couple of areas needing correction/clarification:

1) "Paediatric chordoma is a rare disease; they represent about 5% of all tumours in patients under 20 years of age."This is incorrect and far exceeds incidence.

The sentence has been re-phrased as follows: "Chordomas are a rare disease; they represent 0.2 % of primary brain tumours  and less than 5 % of primary bone tumours . About 5% of all the chordomas tumours occur in patients under 20 years of age."

2) Treatment is noted to be single field optimization, however posterior beam shows heterogeneity across the volume. Please ensure that this is not multi-field optimization.

We confirm the planning technique was a single field optimization (SFO), as reported in the text, not a multi-filed optimization (MFO). In any case it’s worth mentioning as in our SFO approach each field is optimized individually setting not only cost functions for the target coverage (as much uniform as possible),  but also cost functions acting on organs at risks, which leads for instance to the heterogeneities pointed-out by the reviewer. So in that respect the approach cannot be considered as a pure SFUD (single filed uniform dose), but at the same time it’s not an MFO where all the fields are optimized simultaneously.

3) Mixed radiographic response, authors note bony component stable and nodular component decreasing. They both appear to be emanating from the clivus with extraosseous extension or expansion. Is there any thought/rationale to this variable response?

Unfortunately both the components of this lesion (i.e.: the bony-cartilaginous portion and the nodular component) were not object of separate/distinct biopsies, hence we have not any histological reference of double component intra lesion. What is decisive here is the different radiological signal shown by the two components since the diagnosis, which always led to think to a different aggressive behavior: quick growth and response for the nodular component while unchanged dimension for the bony-cartilaginous portion from the diagnosis to the end of the irradiation.    

4) Can authors comment on actual tumor coverage with prescription dose? Presumably the nodular lesion actually received less dose than the other regions given noted dose constraints.

We do agree with the reviewer. We reported the following statements on the text: Such a result is even more meaningfull considering that this tumor area received less than prescribed dose because of its proximity to the spinal cord.  At the same time, despite the dose to the spinal cord was constrained to 54 GyRBE, thanks to the quality of the treatment plan, at least 50% of this tumor area received 70 GyRBE. This future may justify the good radiological outcome.

Reviewer 3 Report

This is a rare case of a pediatric skull base chordoma with an exceptional fast response on proton radiation therapy. The manuscript describes the disease course with a strong emphasis on the radiological response to proton radiation. It is a well-written case report and can be published. 

The pathology description requires more details. Like which stainings have been done? is Brachyury detected in the tissue?  

In addition, more providing more description of the neurological condition of the patient is informative for the readers.

Author Response

Thanks to the reviewer for the interesting suggestions/comments. Herafter we reply point-by-point to his/her questions

Reviewer #3:

This is a rare case of a pediatric skull base chordoma with an exceptional fast response on proton radiation therapy. The manuscript describes the disease course with a strong emphasis on the radiological response to proton radiation. It is a well-written case report and can be published. 

1) The pathology description requires more details. Like which stainings have been done? is Brachyury detected in the tissue?  

The histological picture is of a neoplasm composed by chords and strands of tumor cells embedded in a myxoid background. The tumor cells shows abundant pink cytoplasm and round regular  nuclei with little cytological atypia. Some cells shows multiple cytoplasmic vacuoles creating the classic bubbly appearance of physaliferous cells.

At immunohistochemical examination the neoplasm appear diffusely immunoreactive for Cytokeratin and EMA and shows nuclear immunoreactivity for brachiury and variable  S100 positivity.

The diagnosis of conventional Chordoma was made.

2) In addition, more providing more description of the neurological condition of the patient is informative for the readers.

The patient was in fair psychological and clinical conditions before the start of proton therapy. There were motor and sensory neurological deficits of the following cranial nerves: IX-X-XI-XII. Paralysis of the left vocal cord and difficulty in completely opening the oral cavity were evident as well as severe oropharyngeal dysphagia for solid and liquid foods.

Round 2

Reviewer 1 Report

Thank you for the answers to the queries which I am happy  with.

Reviewer 2 Report

The authors have adequately addressed questions from the initial submission.